# Tailoring Poly(lactic acid) (PLA) Properties: Effect of the Impact Modifiers EE-g-GMA and POE-g-GMA

**DOI:** 10.3390/polym14010136

**Published:** 2021-12-30

**Authors:** Edson Antonio dos Santos Filho, Carlos Bruno Barreto Luna, Danilo Diniz Siqueira, Eduardo da Silva Barbosa Ferreira, Edcleide Maria Araújo

**Affiliations:** Department of Materials Engineering, Federal University of Campina Grande, Campina Grande 58429-900, Brazil; brunobarretodemaufcg@hotmail.com (C.B.B.L.); danilodinizsiqueira@gmail.com (D.D.S.); eduardosbf95@gmail.com (E.d.S.B.F.); edcleidemaraujo@gmail.com (E.M.A.)

**Keywords:** poly(lactic acid), impact modifier, polymer blends, properties

## Abstract

Poly(ethylene-octene) grafted with glycidyl methacrylate (POE-g-GMA) and ethylene elastomeric grafted with glycidyl methacrylate (EE-g-GMA) were used as impact modifiers, aiming for tailoring poly(lactic acid) (PLA) properties. POE-g-GMA and EE-g-GMA was used in a proportion of 5; 7.5 and 10%, considering a good balance of properties for PLA. The PLA/POE-g-GMA and PLA/EE-g-GMA blends were processed in a twin-screw extruder and injection molded. The FTIR spectra indicated interactions between the PLA and the modifiers. The 10% addition of EE-g-GMA and POE-g-GMA promoted significant increases in impact strength, with gains of 108% and 140%, respectively. These acted as heterogeneous nucleating agents in the PLA matrix, generating a higher crystallinity degree for the blends. This impacted to keep the thermal deflection temperature (HDT) and Shore D hardness at the same level as PLA. By thermogravimetry (TG), the blends showed increased thermal stability, suggesting a stabilizing effect of the modifiers POE-g-GMA and EE-g-GMA on the PLA matrix. Scanning electron microscopy (SEM) showed dispersed POE-g-GMA and EE-g-GMA particles, as well as the presence of ligand reinforcing the systems interaction. The PLA properties can be tailored and improved by adding small concentrations of POE-g-GMA and EE-g-GMA. In light of this, new environmentally friendly and semi-biodegradable materials can be manufactured for application in the packaging industry.

## 1. Introduction

With new technologies and new product development, there is more and more concern about the environmental impacts that these materials can cause. In this context, polymeric commodity materials have drawn attention because they are derived from petroleum, since they have high resistance to degradation. As a consequence, they can contribute significantly to the accumulation of waste in natural ecosystems, and thus increase pollution [1,2,3].

Research has advanced toward the development of new eco-friendly materials using “green” technology, targeting materials that favor a closed life cycle, such as biopolymers and biodegradable polymers. Biopolymers are produced from raw materials derived from renewable sources, such as: cellulose, sugar cane, corn and others [4,5,6]. On the other hand, biodegradable polymers are those that undergo degradation from the action of microorganisms in environments that are considered bioactive [7,8].

Poly(lactic acid)—PLA is a biodegradable biopolymer derived from natural resources such as corn, sugar cane and rice. It is an aliphatic polyester thermoplastic and is currently widely used in the processing industry. Regarding the production method, it can be synthesized either by direct condensation polymerization of the lactic acid monomer, or by ring opening of the lactide [9,10,11,12]. PLA has properties similar to crystal polystyrene (PS) and polyethylene terephthalate (PET). In view of this, PLA has aroused interest in the production industry and the scientific community [13,14,15,16]. However, PLA is a very rigid and brittle polymer, which makes it impractical for applications requiring high impact strength. To overcome this problem, the properties of PLA can be tailored by adding impact modifiers.

There are several studies in the literature [17,18,19] on the production of polymer blends by adding natural rubber, polyethylene, ethylene vinyl acetate (EVA) and others into the PLA matrix. Aiming for a higher degree of synergy of mechanical properties, in general, impact modifiers functionalized with glycidyl methacrylate (GMA) are added, aiming at a higher interaction with PLA [20,21]. This may improve the interfacial adhesion between the two phases, leading to higher impact strength and ductility in the system. Therefore, it is desirable to investigate the influence of various impact modifiers that are compatible with PLA, helping to expand the database of “green blends”.

The present investigation aimed to evaluate the influence of elastomeric impact modifiers (POE-g-GMA and EE-g-GMA) addition to the PLA matrix, in order to tailor the thermal, mechanical, thermomechanical properties and the morphology.

## 2. Methodology

### 2.1. Materials

Poly(lactic acid) (PLA) manufactured by NatureWorks, supplied in pellet form by 3D LAB (Betim, Brazil), with a density of 1.24 g/cm^3^ and a flow index (FI) = 6 g/10 min. Elastomeric ethylene graphted with glycidyl methacrylate (EE-g-GMA), supplied by Coace^®^ Plastic (Xiamen, China), FI = 3–8 g/10 min (190 °C/2.16 kg), containing up to 0.8% of GMA. Poly(ethylene octene) grafted with glycidyl methacrylate (POE-g-GMA), supplied by Coace^®^ Plastic (Xiamen, China), FI = 8–16 g/10 min (190 °C/2.16 kg), containing up to 0.8% of GMA. Styrene-(ethylene-butylene)-styrene copolymer (SEBS-MA), FI = 5 g/10 min (230 °C/5 kg), with 1.7% of maleic anhydride, marketed under code FG1901, containing 30% styrene, supplied by Kraton (Houston, Texas, USA). Ethylene methyl acrylate copolymer (EMA), with 24% of methyl acrylate and FI = 7 g/10 min (190 °C/2.16 kg), supplied by Arkema (Colombes Cedex, France), in granule form.

### 2.2. Materials Processing

Prior to the blends preparation, a study of the different impact modifiers influence on PLA was performed. All materials were dried in a vacuum oven at 60 °C for 24 h. Table 1 illustrates the mass proportions (%) of the compositions that were used in the development of the blends. These compositions were developed to analyze the degree of interaction between PLA and functionalized impact modifiers by means of torque rheometry and the Molau test.

After preliminary tests, EE-g-GMA and POE-g-GMA were selected as impact modifiers for PLA, aiming to obtain the binary blends PLA/EE-g-GMA and PLA/POE-g-GMA, with the ratios of 95/5; 92.5/7.5 and 90/10%.

The PLA/EE-g-GMA and PLA/POE-g-GMA blends were dry mixed and later processed in a modular corotating twin-screw extruder, model ZSK (D = 18 mm and L/D = 40), from Coperion Werner & Pfleiderer, with temperature of 170 °C in zones 1 and 2, and 180 °C in the other zones, screw rotation speed of 250 rpm and controlled feed rate of 4 kg/h, with screw profile configured with distributive and dispersive mixing elements (Figure 1). For comparison, the pure PLA was processed under the same conditions as the blends. After processing the systems by extrusion, the materials were granulated and dried in an oven without vacuum for 24 h at a temperature of 60 °C.

The pure PLA and the blends obtained by extrusion were injection molded in an Arburg Allrounder 207C Golden Edition model to obtain specimens. The molding temperature was 180 °C in all zones and the mold temperature was 20 °C.

### 2.3. Characterizations

#### 2.3.1. Torque Rheometry

The rheology curves were obtained in a Thermo Scientific Haake PolyLab QC mixer (Waltham, MA, USA), with roller-type rotors, at 180 °C and rotor speed of 60 rpm, under air atmosphere for 10 min.

#### 2.3.2. Molau Test

The Molau test was performed by dissolving 1 g of PLA and the blends, in 50 mL of *N*-methyl-2-pyrrolidone (NMP), under magnetic stirring at 100 °C. 

#### 2.3.3. Fourier Transform Infrared Spectroscopy (FTIR)

Fourier transform infrared spectroscopy (FTIR) was performed on a BRUKER Vertex 70 Spectrometer (attenuated total reflectance—ATR) (Billerica, MA, USA), in the range 4000 to 400 cm^−1^, with 32 scans and a resolution of 4 cm^−1^.

#### 2.3.4. Impact Strength Test

The Izod impact strength test was performed on notched specimens according to ASTM D256 [23] in a Ceast model Resil 5.5 J device (Turin, Italy), operating with a 2.75 J hammer, at room temperature. The results were analyzed with an average of 10 specimens. 

#### 2.3.5. Shore D Hardness Test

The penetration resistance measurement was carried out according to ASTM D2240 [24], in Shore-Durometer Hardness Type “D” (São Paulo, Brazil) equipment with a 50 N load controlled by springs calibrated using standard indentors for the durometer. The indenter was pressed into the sample for 15 s at five random points on the sample.

#### 2.3.6. Heat Deflection Temperature (HDT)

The heat deflection temperature (HDT) was obtained according to ASTM D648, in a Ceast equipment, model HDT 6 VICAT (Turin, Italy) with a load of 1.82 MPa and heating rate of 120 °C/h (method A). The temperature was determined after the sample was deflected 0.25 mm. The results were analyzed with an average of 3 specimens.

#### 2.3.7. Thermogravimetry (TG)

Thermogravimetry (TG) was obtained in a Shimadzu DTG 60H equipment (Kyoto, Japan), using about 5 mg of sample, heating rate of 10 °C/min and gas flow rate of 100 mL/min, from 30 to 500 °C under nitrogen atmosphere.

#### 2.3.8. Differential Scanning Calorimetry (DSC)

Differential scanning calorimetry (DSC) analysis was performed in a TA Instruments DSC-Q20 equipment (New Castle, DE, USA). The scanning was performed from 30 to 200 °C under heating rate of 10 °C/min, gas flow rate of 50 mL/min and nitrogen atmosphere with a mass of approximately 6 mg. The blends degree of crystallinity was calculated based on the curves obtained in the DSC analyses, according to Equation (1) [25]:(1)$$\%Xc=\frac{\Delta Hm-\Delta Hcc}{\Delta H_{100\%PLA}\times W_{PLA}}$$
where: Δ*H_m_* = melting enthalpy; Δ*H_cc_* = cold crystallization enthalpy; Δ*H*_100%*,PLA*_ = melting enthalpy for 100% crystalline PLA (93,7 J/g) [26]; *W_PLA_* = mass fraction of PLA; *X_c_* = degree of crystallinity.

#### 2.3.9. Scanning Electron Microscopy (SEM)

Scanning electron microscopy (SEM) analyses were performed on the specimen’s fracture surface submitted to the impact test. A scanning electron microscope—VEGAN 3 TESCAN (Brun, Tchéquia)—was used at a voltage of 30 kV under high vacuum. The samples fracture surfaces were coated with gold (sputtering—Shimadzu metallizer—IC 50, using a current of 4 mA for a period of 2 min).

## 3. Results and Discussion

### 3.1. Torque Rheometry

Figure 2 illustrates the torque vs. time curves, as well as the magnification with the stabilized torque for the pure PLA, the impact modifiers and the binary blends, with the fixed ratios of 70/30%.

In Figure 2, it was observed in the first minutes of mixing an initial peak referring to the loading of the material—i.e., when the solid material enters the mixing chamber—that it promotes a resistance to the rotor’s rotation and, as a consequence, there was an increase in torque. The torque then started to reduce, due to the materials plasticization. After approximately 4 min, the curves became constant with small oscillations. This behavior is related to the viscosity stability under the conditions used in this process, i.e., speed of 60 rpm and temperature of 180 °C. It was noted that after 6 min, PLA showed an average torque of 35 N·m, a value similar to SEBS-MA. The impact modifiers EE-g-GMA, POE-g-GMA and EMA, showed lower average torque, 12 N·m, 19 N·m and 15 N·m, respectively. As for the blends, it was observed that there was an increase in the average torque compared to the pure modifiers, with the exception of SEBS-MA, which had a significant reduction. This increase in torque is an indication that there was some interaction between the components. Brito et al. (2016) showed that PLA reacts with polymers functionalized with GMA, generating a reaction between the carboxyl or hydroxyl groups present in PLA and the epoxy group present in GMA.

### 3.2. Molau Test

The Molau test is a fractional dissolution experiment, which is widely used as a qualitative test to indicate whether there has been a reaction between the polymer and the functionalized impact modifiers [27,28]. Therefore, PLA, impact modifiers and blends were dissolved at a ratio of 70/30%.

From this test, it can be evaluated which impact modifiers reacted with PLA, considering that, if there is an interaction between both components, upon dissolution of PLA in the NMP solvent, the modifier is “dragged”, thus changing the solvent color. Figure 3 shows the dissolution of the PLA, impact modifiers and blends in 50 mL volumetric flasks containing the NMP.

Pure PLA was found to dissolve completely in the solvent. The impact modifiers, on the other hand, do not dissolve, as it is possible to clearly observe two distinct phases, indicated in red. It was then possible to notice that the blends containing the modifiers POE-g-GMA and EE-g-GMA showed a single phase with a milky appearance, suggesting indications that there was an interaction between the materials. It can also be observed that the solutions of PLA/EMA and PLA/SEBS-MA blends presented an insoluble part in the solution, which is an indication that there probably were no interactions between the phases [28,29].

### 3.3. Fourier Transform Infrared Spectroscopy (FTIR)

The literature [20,30,31] showed that GMA has an interaction with PLA, confirming the trend verified in torque rheometry and Molau test. Regarding this, the modifiers EE-g-GMA and POE-g-GMA are probably good candidates for tailoring the PLA properties.

Figure 4 shows the infrared spectra of pure PLA, the modifiers EE-g-GMA and POE-g-GMA, and the blends PLA/EE-g-GMA and PLA/POE-g-GMA, in the 70/30% ratios.

The main bands that can be visualized in the spectra are: 750 cm^−1^ for C=O stretching; 865 cm^−1^ for C-COO stretching; 1050 cm^−1^ for C-CH_3_ stretching; 1080 cm^−1^ for C-CH_3_ stretching and -CO stretching (ester); 1185 cm^−1^ for asymmetric CH_3_ stretching and -CO stretching (ester); 1260 cm^−1^ for CH stretching and COC stretching; 1360 cm^−1^ for symmetric CH_3_ stretching and CH bending; 1452 cm^−1^ for asymmetric CH_3_ stretching; 1740 cm^−1^ for the C=O stretching [32,33,34,35]. It was also observed that there were characteristic bands present in EE-g-GMA, POE-g-GMA and the blends at 2860 cm^−1^ referring to the -CH stretching; 2915 cm^−1^ referring to the CH_3_ symmetric stretching and the -CH stretching; 2995 cm^−1^ referring to the CH_3_ asymmetric stretching. This is an indication that there was a chemical reaction between these impact modifiers and PLA [28,36].

Figure 5a,b show the FTIR spectra of the modifiers EE-g-GMA and POE-g-GMA, compared to pure PLA and their respective blends in the 70/30% ratio, in the spectrum between 500 cm^−1^ and 1500 cm^−1^. It can be seen that some bands are similar for all three spectra. However, it is worth considering a low intensity band at 920 cm^−1^, which is typical for GMA (grafted onto both modifiers). This band intensity is related to the GMA grafting degree in the modifiers, which, according to the supplier, is above or equal to 0.8%. It can be seen that there is a decrease in the intensity of this band, thus indicating that a reaction likely occurred during processing and an opening of the epoxy ring, justifying this reduction [33,37].

### 3.4. Impact Strength Test

Figure 6 shows the impact strength of PLA and the blends containing 5; 7.5 and 10% of the impact modifiers EE-g-GMA and POE-g-GMA. This test is important to evaluate the energy dissipation in the material and whether they exhibit a brittle or ductile character.

Literature [38,39,40] showed that the impact strength of PLA can be increased by adding other materials, whether they are polymeric or not. It was observed that pure PLA showed an impact strength of approximately 27 J/m, a value close to the literature [41,42]. With the impact modifiers addition there was an increase in this property. As the content of EE-g-GMA and POE-g-GMA increased, the impact strength increased continuously. Thus, there are indications that there was an improvement in the PLA toughness with the addition of these modifiers, given that the blends show gains of 30, 60 and 108% with the addition of 5; 7.5 and 10% of EE-g-GMA, respectively, while 20, 42 and 140% increased with the addition of 5; 7.5 and 10% of POE-g-GMA, respectively. Apparently, the addition of 10% POE-g-GMA tended to maximize performance under impact, suggesting that there is a critical concentration to Improve the synergy of behavior under impact.

In view of the results presented, EE-g-GMA and POE-g-GMA possibly act by dissipating energy under impact and consequently delaying the propagation of cracks, which are quite common in brittle polymers such as PLA. This behavior was probably due to the impact modifiers elastomeric character used, contributing to an increase of the impact strength of PLA. In addition, it can take into consideration that there is probably an interaction between the GMA functional group and the PLA, generating good interactions between both phases, as verified in the SEM later on. These results are of great importance for green polymer technology, since the addition of a small percentage of the impact modifier already results in an increase in the impact strength of PLA.

### 3.5. Shore D Hardness Test

Figure 7 shows the Shore D hardness of the pure PLA and the blends, with 5; 7.5 and 10% contents of the impact modifiers EE-g-GMA and POE-g-GMA. In this test, the penetration resistance of the material is evaluated.

Pure PLA showed the highest value of hardness, with a value of 75 Shore D due to the high stiffness of this material [43,44,45]. The addition of 5% EE-g-GMA and POE-g-GMA subtly reduced the Shore D hardness compared to pure PLA. However, it was not a significant decrease, since they are within the experimental error. For higher concentrations of impact modifiers (7.5% and 10%), there was a more obvious reduction in Shore D hardness due to greater flexibility, as seen in impact strength.

It was also observed that, as the proportion of modifiers increased, there was also a proportional reduction of these values in Shore D hardness, in view of the increased flexibility and greater amount of material with elastomeric behavior. These results are important for the plastics processing industry, as it increased impact strength and did not significantly compromise penetration resistance. This suggests that new semi-biodegradable materials can be produced for practical applications, contributing to the expanded use of sustainable materials.

### 3.6. Differential Scanning Calorimetry (DSC)

The melting parameters, crystallization and the degree of crystallinity of the pure PLA and the blends are summarized in Table 2. The DSC curves obtained during the second heating and cooling can be seen in Figure 8a,b, respectively. In Figure 8a, it was noted that all compositions showed three main events during heating. The first event is a variation from baseline, which is related to the Tg of PLA, around 62.5 °C [46,47], a value above room temperature, confirming its brittle characteristic, observed in the impact test. The blends PLA/EE-g-GMA and PLA/POE-g-GMA also presented this Tg event, however, without significant variations.

Subsequently, between 110 °C and 120 °C, the samples presented an exothermic peak, referring to cold crystallization (Tcc), an intrinsic phenomenon of PLA. The cold crystallization process originates from the rearrangement of the amorphous region into a crystalline phase. Note that the compositions containing EE-g-GMA show a 7 °C reduction in Tcc, indicating that this impact modifier inhibits this process. Finally, there is an endothermic peak referring to the crystalline fusion, with peak temperature at approximately 152 °C for all compositions. However, it is noted that some compositions showed evidence of double peaks, especially those containing EE-g-GMA. Double peaks form in the presence of crystals of distinct sizes and shapes, melting at different temperatures [48,49].

It can also be observed that during cooling, the composition containing EE-g-GMA showed an exothermic peak related to crystallization at around 70 °C. This can probably be explained by the fact that ethylene crystallizes rapidly, unlike ethylene octene, in which the polymer chain is larger and more difficult to reorganize and crystallize.

From Table 2, it can be observed that there was an increase in the crystallinity of the blends PLA/EE-g-GMA and PLA/POE-g-GMA, compared to pure PLA. Such behavior suggests that the incorporation of low concentration of EE-g-GMA and POE-g-GMA promoted a nucleating effect, contributing to an increase of the crystallinity. Apparently, EE-g-GMA was more effective in enhancing the degree of crystallinity, since it increased by an average of 400% over PLA.

Although POE-g-GMA acted as a nucleating agent, the performance was not comparable to EE-g-GMA. This may be explained by the larger polymer chain size of POE-g-GMA, which hinders crystallization. There was also the presence of the exothermic peak during cooling, as ethylene crystallizes more easily; the higher crystallinity can be attributed to this factor.

### 3.7. Thermogravimetry (TG)

Figure 9 illustrates the TG curves of pure PLA and the blends containing 5; 7.5 and 10% of the impact modifiers EE-g-GMA and POE-g-GMA, and Table 3 presents the data obtained from this analysis.

It is possible to verify that PLA presents a single decomposition step initiated at approximately 300 °C. Further, there was the process of the primary bonds rupture due to the thermal energy, causing the material degradation without presenting residues from 385 °C on. As for the blends, it was observed in the initial degradation temperature Tonset (T_5%_) that there was an increase from 10 to 20 °C. This increase indicates an improvement in the thermal stability of PLA, especially with a higher proportion of impact modifiers. This suggests that there is good interaction between the PLA and the impact modifiers, generating a stabilizing effect, and acting as additives while shifting the stability to a higher temperature. Furthermore, the increase in the degree of crystallinity as verified in the DSC, in general, contributes to improving the thermal stability [50].

According to Table 3, it can also be seen that the blends present a second degradation step around 420 °C, referring to the impact modifiers EE-g-GMA and POE-g-GMA degradation. Thus, it can be inferred that both are more thermally stable than pure PLA. It is also observed that the T_max_, presents a considerable increase of about 70 °C.

Finally, it can be noted that the pure PLA showed no residue at 500 °C, while all the blends showed an average of 3% of the initial mass. This phenomenon indicates that there is an improvement in the thermal stability of PLA due to the chemical and physical interactions between the materials [51]. It is worthwhile to note that in the blends containing EE-g-GMA, there is an increase of the residue as the proportion introduced in the PLA increases. On the other hand, with the introduction of POE-g-GMA, there is a proportional decrease of the residue at 500 °C.

### 3.8. Heat Deflection Temperature (HDT)

Figure 10 shows the HDT behavior of pure PLA and the blends containing 5; 7.5 and 10% of the impact modifiers EE-g-GMA and POE-g-GMA. 

It can be observed that pure PLA presented a value of HDT on the order of 53.6 °C, a value close to that reported in the literature [52,53]. The PLA/EE-g-GMA and PLA/POE-g-GMA blends, regardless of the impact modifier content, showed no significant variations compared to pure PLA. Such behavior suggests that the addition of a GMA functionalized impact modifier in low concentration did not promote a deleterious effect on the thermomechanical strength in the PLA matrix.

From a technological point of view, the results are important for new eco-friendly materials production and development, with a good balance of properties, especially impact strength and HDT. Regarding the blends, EE-g-GMA and POE-g-GMA provided comparable values in HDT, considering that the values are within the experimental error.

### 3.9. Scanning Electron Microscopy (SEM)

Figure 11 shows the SEM images of the pure PLA and the blends samples containing 5; 7.5 and 10% of the impact modifiers EE-g-GMA and POE-g-GMA, at a magnification of 5000×, after being subjected to the impact strength test.

From the micrographs, the phase dispersion and interfacial adhesion can be analyzed. The pure PLA exhibits a morphology with smooth lines and without plastic deformation, characteristic of amorphous and exhibiting brittle fracture polymers [46,54,55], as can be seen in Figure 11a. In Figure 11b.1–b.3, are the blends containing 5; 7.5 and 10% EE-g-GMA, respectively, and Figure 11c.1–c.3 the blends containing 5; 7.5 and 10% POE-g-GMA, respectively.

Figure 11 shows that the PLA/EE-g-GMA and PLA/POE-g-GMA blends showed phase separation, where EE-g-GMA and POE-g-GMA particles are dispersed in the PLA matrix. Clearly, there was the presence of well adhered EE-g-GMA and POE-g-GMA particles in the PLA matrix, justifying the increases in impact strength. However, it was also noted that the morphology showed some voids, indicating that some particles were pulled out during the impact test. At the same time, it can be seen that increasing the content of the modifiers EE-g-GMA and POE-g-GMA caused an increase in the average particle size, especially at 10%, which can be attributed to the coalescence phenomenon [56]. However, it can be observed that the dispersion of both modifiers, EE-g-GMA and POE-g-GMA, was uniform in the PLA matrix, contributing to increase the energy dissipation level. Furthermore, some fibrils were noted in the PLA/EE-g-GMA and PLA/POE-g-GMA blends, as well as whitish zones, characteristic of a plastic deformation, and corroborating with the impact strength results.

## 4. Conclusions

The mechanical, thermal, thermomechanical properties and morphology of semi-biodegradable blends of PLA with EE-g-GMA and POE-g-GMA were investigated. It was found that the properties of PLA can be tailored by adding small concentrations of EE-g-GMA and POE-g-GMA, generating promising eco-friendly materials. The blends PLA/EE-g-GMA and PLA/POE-g-GMA showed better impact properties and thermal stability, compared to pure PLA. At the same time, HDT was not affected, due to the nucleating effect of the impact modifiers EE-g-GMA and POE-g-GMA in the PLA matrix. As a consequence, the increase in crystallinity contributed to maintaining the thermomechanical strength (HDT), Shore D hardness, and shifting the thermal stability to higher temperature of the PLA/EE-g-GMA and PLA/POE-g-GMA blends. The morphology obtained by SEM suggested a good interaction between PLA and the EE-g-GMA and POE-g-GMA systems, due to the glycidyl methacrylate functional group. In general, new ecoproducts can be manufactured with the PLA/EE-g-GMA and PLA/POE-g-GMA blends, aiming at applications in the packaging industry. These blends are less polluting to the environment and non-toxic, contributing to sustainable development.

## Figures and Tables

**Figure 1 polymers-14-00136-f001:**
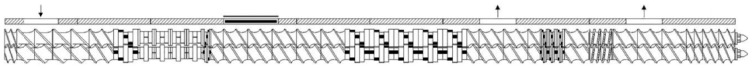
Extruder screw profile used with distributive and dispersive mixing elements [22].

**Figure 2 polymers-14-00136-f002:**
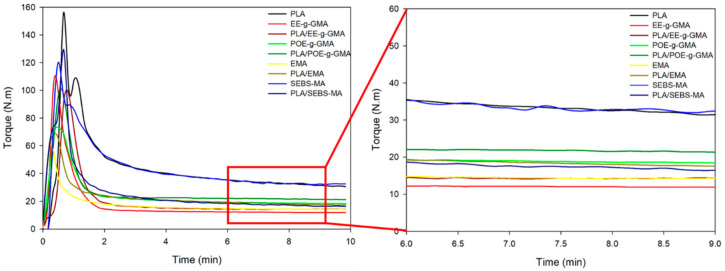
Torque versus time curves of pure PLA, impact modifiers and blends (70/30% by mass).

**Figure 3 polymers-14-00136-f003:**
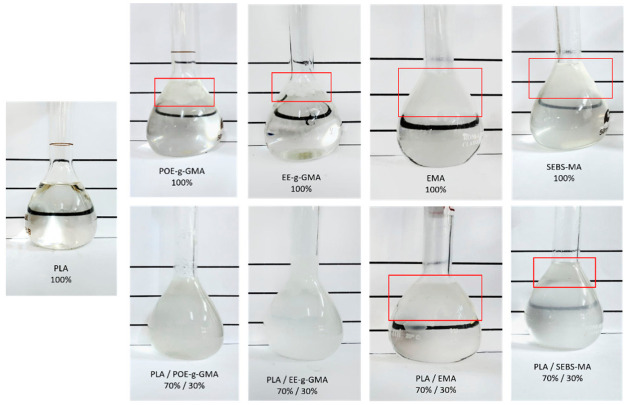
Dissolution of pure PLA, impact modifiers and blends (70/30%) in NMP solvent.

**Figure 4 polymers-14-00136-f004:**
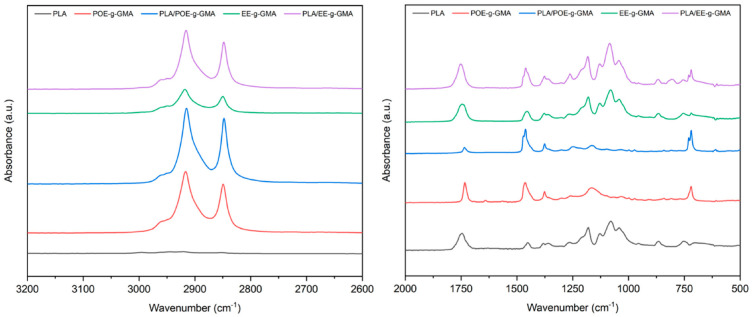
Infrared spectra of pure PLA, EE-g-GMA, POE-g-GMA and the blends (70/30%).

**Figure 5 polymers-14-00136-f005:**
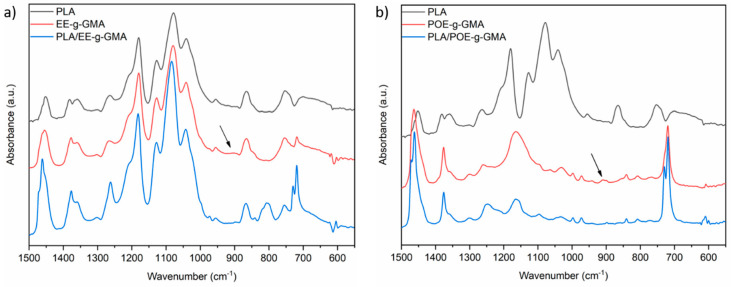
Infrared spectra of pure PLA, (**a**) EE-g-GMA and of the blend PLA/EE-g-GMA and (**b**) POE-g-GMA and of the blend PLA/POE-g-GMA.

**Figure 6 polymers-14-00136-f006:**
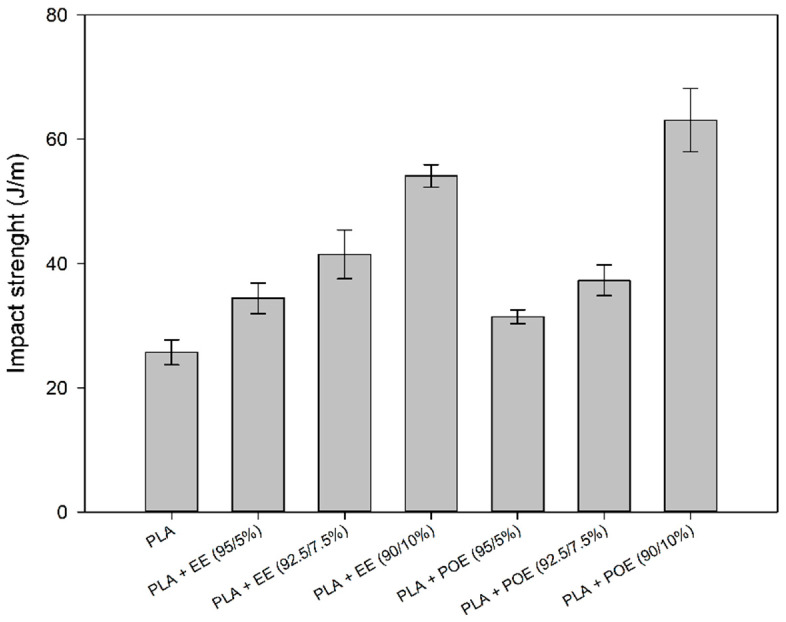
Impact strength of pure PLA and blends containing 5; 7.5 and 10% EE-g-GMA and POE-g-GMA.

**Figure 7 polymers-14-00136-f007:**
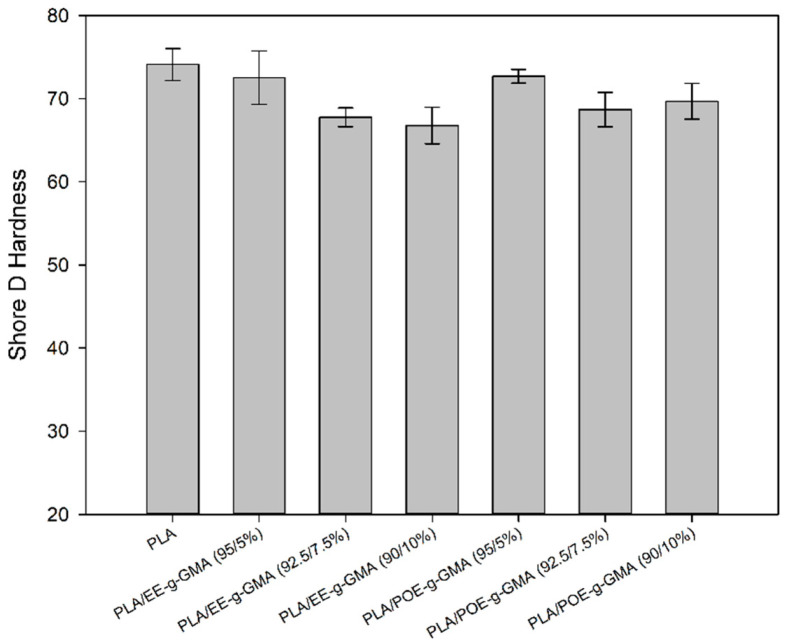
Shore D hardness of pure PLA and the blends containing 5; 7.5 and 10% 5; 7.5 and 10% EE-g-GMA and POE-g-GMA.

**Figure 8 polymers-14-00136-f008:**
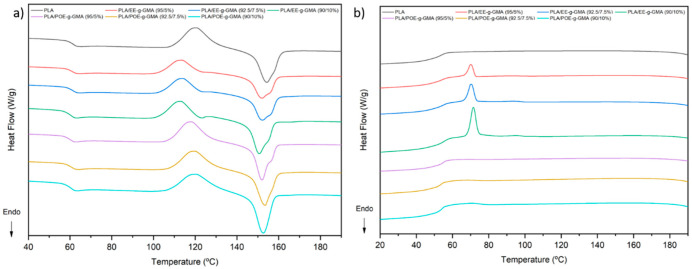
DSC curves of pure PLA and the blends containing 5; 7.5 and 10% EE-g-GMA and POE-g-GMA, (**a**) heating and (**b**) cooling.

**Figure 9 polymers-14-00136-f009:**
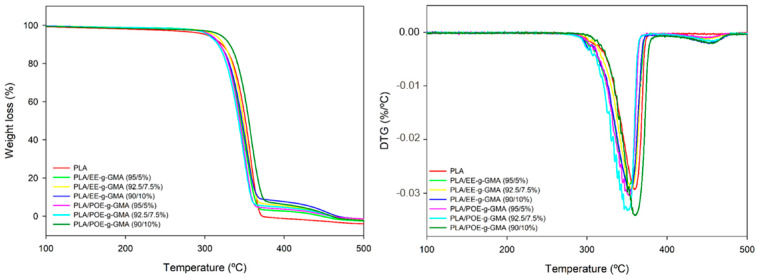
TG and DTG curves of pure PLA and the blends containing 5; 7.5 and 10% 5; 7.5 and 10% EE-g-GMA and POE-g-GMA.

**Figure 10 polymers-14-00136-f010:**
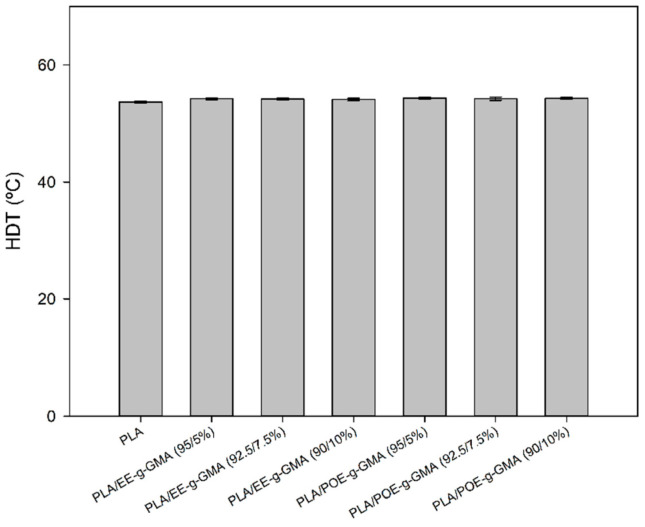
Heat deflection temperature of pure PLA and blends containing 5; 7.5 and 10% of EE-g-GMA and POE-g-GMA.

**Figure 11 polymers-14-00136-f011:**
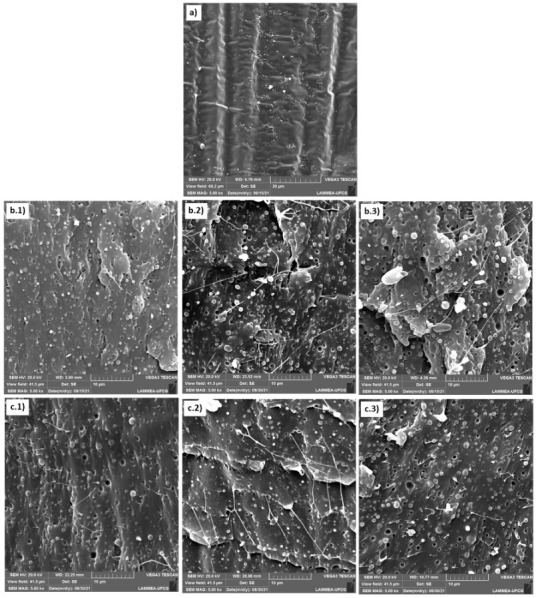
Micrographs obtained by SEM after impact strength test of: (**a**) pure PLA, (**b**) PLA/EE-g-GMA and (**c**) PLA/POE-g-GMA, in the proportions of: (1) 5%; (2) 7.5% and (3) 10%.

**Table 1 polymers-14-00136-t001:** Obtained systems compositions.

Materials	PLA (%)	EE-g-GMA (%)	POE-g-GMA (%)	SEBS-MA (%)	EMA (%)
**PLA**	100	-	-	-	-
**PLA/EE-g-GMA**	70	30	-	-	-
**PLA/POE-g-GMA**	70	-	30	-	-
**PLA/SEBS-MA**	70	-	-	30	-
**PLA/EMA**	70	-	-	-	30

**Table 2 polymers-14-00136-t002:** Data obtained in the DSC of the compositions.

	Tg (°C)	Tcc (°C)	ΔHcc	Tm (°C)	ΔHm	Tc (°C)	Xc (%)
**PLA**	63.3	120.6	21.75	154.3	22.61	-	0.92
**PLA/EE-g-GMA (95/5%)**	62.7	113.3	10.80	152.0	15.12	70.4	4.85
**PLA/EE-g-GMA (92.5/7.5%)**	63.2	113.9	13.77	152.2	18.25	70.3	5.17
**PLA/EE-g-GMA (90/10%)**	62.4	113.1	14.76	150.8	18.58	71.5	4.53
**PLA/POE-g-GMA (95/5%)**	61.8	118.1	20.53	152.0	22.10	-	1.76
**PLA/POE-g-GMA (92.5/7.5%)**	62.7	119.7	22.56	153.5	23.76	-	1.38
**PLA/POE-g-GMA (90/10%)**	62.5	120.3	20.06	152.7	22.53	-	2.93

Tg: Glass temperature transition; Tcc: Cold crystallization temperature; ΔHcc: Cold crystallization enthalpy; Tm: Melting temperature; ΔHm: Melting enthalpy; Tc: Crystallization temperature; Xc: Crystallization degree.

**Table 3 polymers-14-00136-t003:** Degradation data obtained from TG of the compositions.

	T_5%_ (°C)	T_50%_ (°C)	T_max_ (°C)	Residue at 500 °C (%)
**PLA**	295.9	348.5	370.2	0.00
**PLA/EE-g-GMA (95/5%)**	306.4	345.5	435.9	2.24
**PLA/EE-g-GMA (92.5/7.5%)**	312.0	348.8	452.4	3.07
**PLA/EE-g-GMA (95/10%)**	310.0	346.8	459.8	3.27
**PLA/POE-g-GMA (95/5%)**	303.9	344.1	447.2	3.25
**PLA/POE-g-GMA (92.5/7.5%)**	306.7	342.2	451.0	2.56
**PLA/POE-g-GMA (95/10%)**	318.8	355.0	453.1	2.46

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
