# Peer review of "Tailoring Poly(lactic acid) (PLA) Properties: Effect of the Impact Modifiers EE-g-GMA and POE-g-GMA"

_polymers, 2021, doi:10.3390/polym14010136_

Round 1

Reviewer 1 Report

The topic of the manuscript is very interesting, novel and it provides new information to the scientific field. However, I propose the following changes for the improvement of the manuscript quality.

I have a fundamental problem with understanding what blends were used, because in Table 1 there is neither EE-g-G-GMA nor POE-g-G-GMA ... And throughout the manuscript, Abstract, Introduction ... they are mentioned. Are these maybe EE-g-GMA and POE-g-GMA, because such abbreviations also appear in the title and also in the text? If I am wrong and these are different materials, then how do I understand: the letter -g- comes from graft, and what does -G- come from? Please refer to this very important and fundamental issue. In my opinion, this is probably an editorial error and an additional letter -G-, which causes big problems after all.

Line 52, expand the EVA abbreviation;

Line 65, please check FI unit;

Please check all units in the manuscript.

Line 96, 268, 272, 275, 277 ... please use the correct symbol for degrees;

Line 104 - it seems to me that according to IUPAC "Organic compounds nomenclature", N- should be written in italics;

Line 111, 114 - please provide references to standards;

Eq. 1 - maybe a multiplication sign should be inserted;

Fig. 2, 5, 9 - please improve the quality; sometimes too many lines overlap and make data difficult to interpret;

Fig. 6, 7, 10 - please think about the better captions of the bars - they take 1/3 of the size of the whole Fig.

Line 156 - there seems to be an extra space at the beginning of the sentence;

Line 172 - I'd rather just use: in the NMP solvent;

Table 2.- I would suggest inserting an explanation of the symbols (below the table)

Table 3. - please correct the Residuo a 500 ... - the language of the journal is English;

The issue of the use of Portuguese in the manuscript is also repeated in lines 356-361. The reviewer is not fluent in this language, but notes that this paragraph appears to be a repetition of the previous one (lines 350-355);

Fig. 11 - please place SEM micrographs of larger sizes and of much better quality. I would also suggest changing the way of signing because the black font in the dark image is not very visible and legible;

All things considered, this manuscript addresses an important research topic, however there are some mistakes, which should be corrected. Therefore, I suggest the major revision of this article.

Author Response

Dear reviewer, 

Reviewer 2 Report

The manuscript describes mechanical, thermal, thermomechanical properties and morphology of blends of PLA modified with EE-g-GMA and POE-g-GMA. These blends could be less polluting to the environment and non-toxic, contributing to sustainable development. The research is interesting in polymer field and the manuscript could be considered for publication after the revision.

- Poly(ethylene-octene) grafted with glycidyl methacrylate (POE-g-GMA) and ethylene elastomeric grafted with glycidyl methacrylate (EE-g-GMA) were used as impact modifiers for the PLA. Authors should clearly discover why the modifiers have been chosen from various other modifiers ?

- The modifiers were used in a proportion of 5; 7.5 and 10%. How it was decided to use these proportions ?

- Blends of the modifiers were processed in a twin-screw extruder and injection molded. The two methods of preparation are connected with additional costs.  Is it not possible to prepare suitable blend by using just one of the mentioned methods ?

- The 10% addition of EE-g-G-GMA and POE-g-GMA promoted significant increases in impact strength of PLA. What would be influence if more than 10% of the EE-g-G-GMA and POE-g-GMA would be used ?

- By thermogravimetry (TG), the prepared blends showed increased thermal stability. What would be a mechanism of the increased stability.

- It is mentioned that from the developed product “new environmentally friendly and semi-biodegradable materials can be manufactured for application in the packaging industry.”  Could the properties of the mentioned environmentally friendly and semi-biodegradable materials demonstrated experimentaly in the manuscript ?

Author Response

Dear reviewer,

Round 2

Reviewer 1 Report

My suggestions were taken into account and in my opinion the article may be published in this form after the Editor's decision.

Kind regards

Reviewer 2 Report

I think that after the revision Editor could consider the paper for publication.

Best regards,